# Bilateral Adaptive Cross-Modal Fusion Prompt Learning for CLIP

Submission Id: 3059

## ABSTRACT

In the realm of CLIP adaptation through prompt learning, it is important to emphasize the pivotal role that the proper alignment of visual and textual representations plays when adapting the CLIP to downstream tasks. We propose that the proper alignment for downstream tasks is determined by the **flexibility** of the interaction between cross-modal information, which compensates for the absence of contrastive loss during the adaptation process. However, the current prompt learning methods, such as isolated modifications to the visual or language branches of CLIP or the employment of uni-directional cross-modal fusion, are not sufficient to explore the full potential of the mutual interaction between visual and textual modalities. To overcome this limitation, we propose a new paradigm for the CLIP prompt learning community, named **Bil**ateral Adaptive Cr**o**ss-Modal Fusi**o**n Pr**o**mpt Learning (*Bloom*) which includes two enhancements. First, we propose using projection functions for bi-directional modality transformation and fusion functions to encourage the mutual interaction between corresponding layers within both the image and text encoders. Second, we propose an adaptive manner that automatically searches the optimal combination of cross-modal information at each layer. These two improvements ensure a more efficient and flexible integration of the two modalities, thereby achieving proper alignment for specific downstream tasks. We put our method to the test in terms of base-to-novel, cross-dataset, and cross-domain evaluations on 15 image classification datasets. The results demonstrate a significant performance enhancement achieved by *Bloom*.

## CCS CONCEPTS

• **Computing methodologies** → *Scene understanding*.

## KEYWORDS

Adaptive Cross-modal Fusion, Prompt Learning, CLIP

## 1 INTRODUCTION

The fundamental vision-language pre-trained models, particularly the groundbreaking CLIP model [20], have demonstrated extraordinary generalization capabilities across a wide array of downstream computer vision tasks. Examples of these tasks include but are not limited to, image classification [10, 12, 29, 30], semantic segmentation [6, 15, 21, 31], object detection [4, 14, 28], and action recognition [11, 17, 25, 26]. The primary objective of CLIP is to

**Unpublished working draft. Not for distribution.**

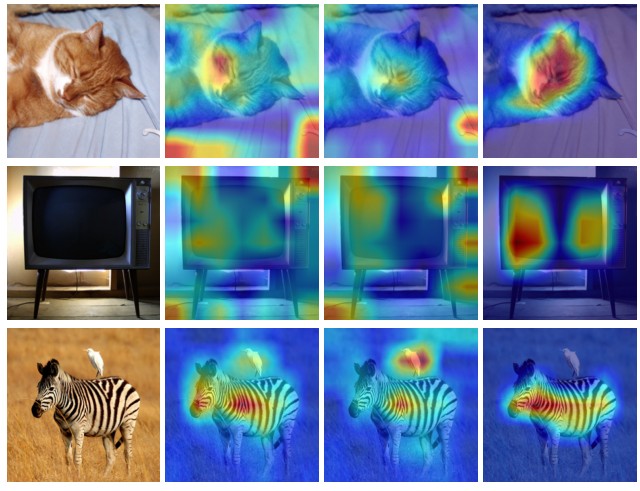

(a) Origin  (b) CLIP ViT-B/16  (c) CoOp ViT-B/16  (d) Ours ViT-B/16

**Figure 1: Comparison of gradient class attention maps for ground-truth category. Both CLIP [20] and the prompt learning method CoOp [30] fail as they focus on irrelevant regions, which results from the misalignment between visual and textual representations.** *Bloom* **effectively addresses it leveraging sufficient flexibility of cross-modal interaction.**

effectively align language and vision representations into a unified feature space, utilizing an extensive web-scale dataset comprising approximately 400 million image-text description pairs. In contrast to the conventional approach of fine-tuning the entire model, the prevailing research methodology primarily leverages prompt learning, which is inspired by advancements in the field of natural language processing (NLP), to adapt the capabilities of CLIP to downstream tasks.

Current prompt learning methodologies can be broadly classified into two distinct categories. The first category encompasses uni-modal prompt learning techniques [10, 29, 30], which exclusively adjusts either the language or vision branch of the CLIP model, as shown in Fig. 2 (a). The second category is multimodal prompt learning (MaPLe) [12], which simultaneously adjusts both language and vision branches, as shown in Fig. 2 (b). MaPLe incorporates a learnable textual prompt in conjunction with a layer-wise coupling function, such as a single layer MLP, to facilitate language-to-vision ($L \rightarrow V$) transformation and generate a visual prompt.

Despite achieving a certain level of efficiency, the prompt learning adaptation process consistently exhibits a distinct objective function compared to the pre-training process. For example, single-label image classification tasks employ CrossEntropy loss for training, whereas contrastive loss is absent. This difference in optimization objectives inevitably disrupts the original alignment between language and visual representations. In Fig. 1, we analyze the failure

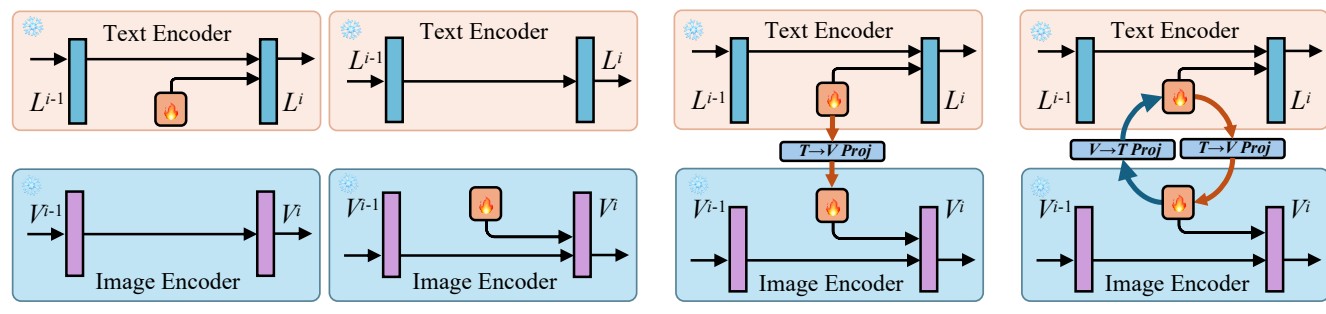

(a) Uni-modal prompt learning      (b) Uni-directional cross-modal fusion      (c) Bilateral cross-modal fusion

**Figure 2: Comparison of (a) Uni-modal prompt learning, (b) Uni-directional cross-modal fusion and our proposed (c) Bilateral cross-modal fusion strategy. 🔥 represents trainable parameters, ❄️ represents frozen parameters.**

cases of CLIP and prompt learning CoOp by examining the gradient class attention maps for the ground-truth category (i.e., cat, TV, and zebra, respectively). The misalignment of the two modalities causes both CLIP and CoOp to concentrate on irrelevant regions. As a result, an efficient prompt learning method should strive to establish proper alignment for a specific downstream task. In this paper, we believe that the proper alignment is determined by the *flexibility* of cross-modal information interaction. However, upon reviewing existing prompt learning methods, we find that uni-modal prompt learning inadvertently overlooks the multi-modal aligned structure inherent to the CLIP model. As a result, solely adjusting the vision or language branch proves inadequate for the effective adaptation of CLIP. Although MaPLe has recognized the importance of cross-modal interaction, the uni-directional prompt transformation remains insufficient. Consequently, a key question arises: *How can we make prompt learning as flexible as possible?*

To that end, we propose a new paradigm for prompt learning community, named **Bil**ateral Adaptive Cr**o**ss-Modal Fusi**o**n Pro**m**pt Learning (*Bloom*) with two enhancements to address these shortcomings. First, we ensure a bi-directional interaction between the two modalities. Specifically, as shown in Fig. 2 (c), we simultaneously initialize individual vision and language prompts for the corresponding branches at each layer. Subsequently, we employ two bottleneck-style projection functions to achieve *language-to-vision* ($L \rightarrow V$) and *vision-to-language* ($V \rightarrow L$) prompt transformations. Second, we use a fusion function with an adaptive strategy to attain cross-modal fusion in an end-to-end manner, which allows *Bloom* to automatically search an optimal convex combination of bilateral cross-modal information at each layer, thereby ensuring proper alignment.

We conduct a comprehensive experimental evaluation on 15 image classification datasets to validate the generalization capabilities of the *Bloom* in terms of base-to-novel generalization, cross-dataset, and cross-domain generalization. The results provide evidence that *Bloom* effectively enhances the prompt learning of the CLIP model by improving the generalization performance for unseen classes, as well as in varying data distribution and domain scenarios.

Our contributions can be summarized in three aspects:

- We propose a new **Bil**ateral Adaptive Cr**o**ss-Modal Fusi**o**n Pro**m**pt Learning (*Bloom*) paradigm for prompt learning

that explores **flexible** cross-modal interactions to attain appropriate alignment for specific downstream tasks.
- We propose an adaptive cross-modal fusion function to ensure our proposed *Bloom* to automatically search the optimal combination of cross-modal information, which further enhances the flexibility of prompt learning.
- Through extensive experimentation, we demonstrate that *Bloom* significantly advances the current state of multi-modal prompt learning, achieving new state-of-the-art results across 15 image classification datasets.

## 2 RELATED WORK

**Contrastive Language-Image Pre-training.** Contrasting the traditional approach in the computer vision field, which relies on pre-training using manually annotated datasets, CLIP [20] introduces a web-scale image-text pair dataset for noisy contrastive learning. This approach empowers the model to acquire a broad spectrum of computer vision tasks during the pre-training phase. CLIP showcases two notable strengths. Firstly, it exhibits exceptional representation learning capabilities, achieving linear-prob performance on par with fully supervised training models. For example, the zero-shot classification results of CLIP outperform the fully supervised ResNet-50 in 16 out of 27 image classification datasets, including ImageNet [3]. Secondly, CLIP displays enhanced robustness against natural distribution shifts, making it more adaptable and resilient to varying domains. As a multifaceted cross-modality foundation model, CLIP is extensively employed across a diverse range of downstream tasks such as image classification [10, 12, 29, 30], object detection [4, 14, 28], semantic segmentation [6, 15, 21, 31], action recognition [11, 17, 25, 26].

**Prompts Learning for CLIP.** The key to adapting CLIP lies in narrowing the gap between downstream tasks and pre-training tasks. There are two strategies. The first, represented by fully fine-tuning, involves adjusting the parameters of CLIP. However, this approach not only entails substantial training costs but also undermines the zero-shot capabilities of CLIP. The second strategy, known as Prompt Learning, bridges the gap between downstream task data and pre-training data. The foundation model remains unaltered and retains the robustness and zero-shot capabilities inherent in CLIP. Prompt learning methods can be broadly categorized into two

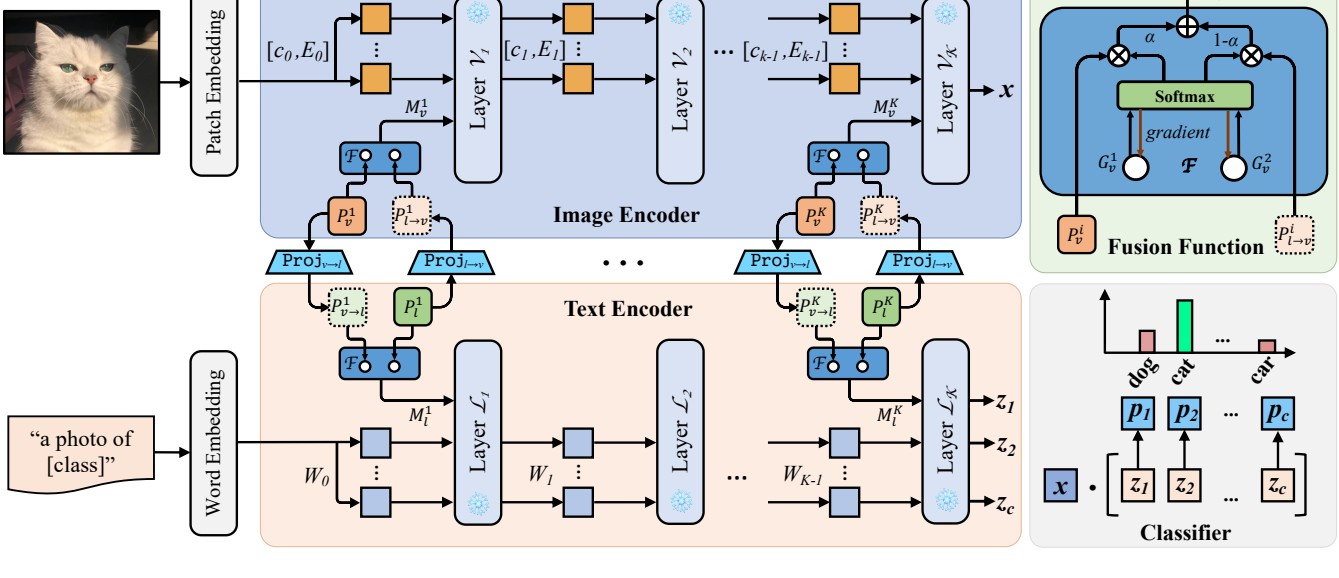

**Figure 3: Framework of bilateral adaptive cross-modal fusion prompt learning (*Bloom*). *Bloom* incorporates a bi-directional modality transformation through the utilization of two projection functions and subsequently achieves cross-modal fusion via fusion functions. Consequently, *Bloom* facilitates a mutual interaction between the two modalities and proper alignment.**

main streams. First, uni-modal prompts focus on learning either textual or visual prompts for CLIP. For instance, CoOp [30] aims to learn prompts for the language branch of CLIP, while CoCoOp [29] incorporates instance-dependent bias as a condition to guide the learning of prompts, thereby enhancing the generalization ability to unseen classes. Additionally, [26] introduces motion-aware prompts to adapt CLIP for action recognition tasks. VPT [10] integrates a small number of learnable parameters, which are then prepended to the input sequence of each Transformer layer. Second, multimodal prompt [12] involves learning prompts for both language and vision branches to improve the alignment between visual and textual representations.

## 3 REVISITING CLIP

Our method is built upon the Transformer-based CLIP [20], which integrates both a *text encoder* and an *image encoder* with the primary objective of generating textual and visual representations, respectively. To effectively understand our method, it is important to revisit the structure and functionality of the CLIP model.

### 3.1 Image Encoder

The image encoder, denoted as $\mathcal{V}$, is composed of $K$ transformer layers. Given an RGB image as input, we initially partition the image into $N$ non-overlapping patches, subsequently projecting these image patches into image embeddings $E_0 = [e_0^0, e_0^1, \cdots, e_0^N] \in \mathbb{R}^{N \times d_v}$. From the perspective of the $i^{th}$ layer within $\mathcal{V}$, the image embeddings $E_{i-1}$ serve as input to $\mathcal{V}_i$, accompanied by a learnable class token $c_{i-1}$. This class token is instrumental in capturing global contextual information. This calculation can be represented as:

$$[\, c_i,\ E_i\, ] = \mathcal{V}_i([\, c_{i-1},\ E_{i-1}\, ]), \tag{1}$$

where $[\cdot, \cdot]$ denotes concatenate operation. Lastly, the class token $c_K$ is projected by $\texttt{ImageProj}(\cdot)$ to obtain the final image representation:

$$x = \texttt{ImageProj}(c_K),\ c_K \in \mathbb{R}^{d_v}. \tag{2}$$

### 3.2 Text Encoder

The text encoder, denoted as $\mathcal{L}$, features a layer count that is identical to that of the image encoder, $\mathcal{V}$. The textual description corresponding to the input image undergoes an initial projection to text embeddings $W_0 = [w_0^0,\ w_0^1,\ \cdots,\ w_0^H] \in \mathbb{R}^{H \times d_l}$ via the $\texttt{Tokenizing}$ process. In this context, $H$ is typically set to 77. With respect to the $i^{th}$ layer within $\mathcal{L}$, the text embeddings $W_{i-1}$ are utilized as input to $\mathcal{L}_i$, namely that:

$$W_i = \mathcal{L}_i(W_{i-1}),\ i = 1,\ 2,\ \cdots,\ K. \tag{3}$$

At last, the final text representation $z$ is obtained by projecting the text embeddings corresponding to the last token of the $K^{th}$ transformer layer via $\texttt{TextProj}(\cdot)$:

$$z = \texttt{TextProj}(w_K^H),\ w_K^H \in \mathbb{R}^{d_l}. \tag{4}$$

### 3.3 Zero-shot Prediction

The CLIP model is remarkably well-suited for zero-shot prediction tasks. Given an image and a set of corresponding $C$ classes, we first extract the image representation $x$ by leveraging the image encoder. Subsequently, we construct $C$ descriptions based on a hand-crafted prompt, such as "a photo of a [CLS].", and obtain the corresponding text representations $[z_0,\ z_i,\ \cdots,\ z_C]$. Finally, we derive the ultimate prediction $\hat{y}$ by selecting the category exhibiting the highest cosine similarity, which is computed using the Softmax

function with an accompanying temperature coefficient $\tau$:

$$p(\hat{y}|x) = \frac{\exp(\text{sim}(x, z_{\hat{y}})/\tau)}{\sum_{j=1}^{C} \exp(\text{sim}(x, z_j)/\tau)}. \tag{5}$$

# 4 BILATERAL ADAPTIVE CROSS-MODAL FUSION PROMPT LEARNING

Prior research [12] has determined that uni-modal adaptation [10, 29, 30], which involves exclusively learning prompts for either the vision or language branch, results in a sub-optimal solution. This is due to the degradation of CLIP's flexibility in aligning textual and visual representations. Consequently, the primary principle for an optimal prompt learning methodology is to establish a proper alignment for specific downstream tasks. Although MaPLe [12] proposes the first multimodal prompt learning method to investigate cross-modal interaction via a uni-directional coupling function, such as a single layer of MLP, it still remains insufficient in terms of the flexibility. In this section, we first bridge the image and text encoders at each layer through a bi-directional transformation which consists of two light-weight projectors. Then we introduce the adaptive cross-modal fusion function in details.

## 4.1 Bilateral Cross-modal Fusion

We initially propose a comprehensive framework designed to facilitate bi-directional fusion across multimodal prompts, as illustrated in Fig. 3. Specifically, at each Transformer layer, a visual prompt $P_v$ and a textual prompt $P_l$ are concurrently assigned to their respective branches. Both prompts comprise a series of learnable vectors, with the vector length $L$ being consistent across $P_v$ and $P_l$.

We utilize distinct methods to initialize the visual and the textual prompts. In particular, textual prompts are initialized using handcrafted prompts, such as "a photo of a [CLS].", while visual prompts are established through a Gaussian distribution $\mathcal{N}(0, 0.2)$. Formally, this can be expressed as:

$$\begin{aligned} P_v &= \{p_v^j \in \mathbb{R}^{d_v}\}_{j=1}^L, \\ P_l &= \{p_l^j \in \mathbb{R}^{d_l}\}_{j=1}^L \sim \mathcal{N}(0, 0.2), \end{aligned} \tag{6}$$

where $d_v$ and $d_l$ denote the dimensions of visual and textual representations, respectively.

**Projection Functions.** To facilitate the transformations between vision and language branches, a pair of projection functions is employed, symbolized as $\text{Proj}_{l\to v}(\cdot)$ and $\text{Proj}_{l\to v}(\cdot)$. The projection functions serve a dual purpose: they facilitate cross-modal transformations and simultaneously align the dimensions of the prompts, thereby enabling the implementation of a cross-modal fusion strategy during the *fusion* stage.

A straightforward approach would be to employ a single MLP, analogous to the coupling function of MaPLe. However, the considerable number of trainable parameters adversely impacts efficiency. As a result, we design a bottleneck-style projection function. For instance, in the case of $\text{Proj}_{v\to l}(\cdot)$, the initial reduction layer $S_{v\to l}^r$ compresses the dimension of $P_v$ from $d_v$ to $d_m$, where $d_m$ is significantly smaller than $d_v$. This reduction layer is followed by a non-linear ReLU and an expansion layer $S_{v\to l}^e$, which increases the dimension from $d_m$ to $d_l$. At this point, the $V \to L$ transformation is complete and vice versa. As a result, the trainable parameters of

$\text{Proj}_{v\to l}(\cdot)$ or $\text{Proj}_{l\to v}(\cdot)$ is merely $d_m \times (d_v + d_l)$, which is much smaller than the one of MaPLe, i.e, $d_v \times d_l$. The computation of projection functions can be represented as:

$$\begin{aligned} \text{Proj}_{v\to l}(P_v) &= S_{v\to l}^e \circ \text{ReLU} \circ S_{v\to l}^r, \\ \text{Proj}_{l\to v}(P_l) &= S_{l\to v}^e \circ \text{ReLU} \circ S_{l\to v}^r. \end{aligned} \tag{7}$$

Finally, these functions generate two transformed prompts, called $P_{v\to l}$ and $P_{l\to v}$, namely that:

$$\begin{aligned} P_{v\to l} &= \text{Proj}_{v\to l}(P_v) \in \mathbb{R}^{L \times d_l}, \\ P_{l\to v} &= \text{Proj}_{l\to v}(P_l) \in \mathbb{R}^{L \times d_v}. \end{aligned} \tag{8}$$

Consequently, we propose employing a fusion function, denoted as $\mathcal{F}(\cdot, \cdot)$, to generate a synthesis of cross-modal prompts, ultimately achieving mutual interaction between the two modalities. This process can be mathematically represented as follows:

$$\begin{aligned} M_v &= \mathcal{F}(P_v, P_{l\to v}), \\ M_l &= \mathcal{F}(P_l, P_{v\to l}). \end{aligned} \tag{9}$$

Here, $M_v$ and $M_l$ symbolize the fused prompts, which encapsulate information derived from both visual and language branches, effectively bridging the gap between the two modalities and enhancing the overall performance.

## 4.2 Adaptive Cross-modal Fusion Function

A prevalent approach for cross-modal fusion involves the direct addition of bilateral cross-modal information. However, this method remains inflexible, as it presumes uniform interaction information sharing across every layer between the two modalities. Our adaptive cross-modal fusion strategy is predicated on a hypothesis that the adaptation process necessitates varying degrees of additional information from other modalities at different layers. Consequently, we propose an adaptive strategy designed to enable prompt learning to autonomously search for an optimal combination of cross-modal information, thereby offering maximum flexibility.

As shown in Fig. 3, we introduce a learnable gate vector, $\mathcal{G}_v \in \mathbb{R}^2$, for the vision branch, and another learnable gate vector, $\mathcal{G}_l \in \mathbb{R}^2$, for the language branch within each layer. These gate vectors are initialized by uniform distribution, while produce binomial distributions during training, generating convex combinations of multimodal prompts and allowing for flexibility in the search for optimal solutions. This gating function, termed $\mathcal{F}_g$, can be represented as:

$$\begin{aligned} \mathcal{F}_g(P_v, P_{l\to v}) &= \text{softmax}(\mathcal{G}_v) \cdot \begin{pmatrix} P_v \\ P_{l\to v} \end{pmatrix}, \\ \mathcal{F}_g(P_l, P_{v\to l}) &= \text{softmax}(\mathcal{G}_l) \cdot \begin{pmatrix} P_l \\ P_{v\to l} \end{pmatrix}. \end{aligned} \tag{10}$$

## 4.3 The Usage of Fused Prompts

The mixture of cross-modal prompts is subsequently appended to the corresponding image or text tokens. Specifically, at the $i^{th}$ layer $\mathcal{V}_i$ within the image encoder, $M_v^i$ is first concatenated with the image embeddings $E_{i-1}$ and the class token $c_{i-1}$ from the previous layer. This concatenated output is then fed into $\mathcal{V}_i$ to facilitate $L \to V$ interaction, namely:

$$[c_i, E_i, \_] = \mathcal{V}_i([c_{i-1}, E_{i-1}, M_v^i]), \tag{11}$$

**(a) Average over 11 datasets**

|  | Base | Novel | HM |
|---|---|---|---|
| CLIP [20] | 69.34 | 74.22 | 71.70 |
| CoOp [30] | 82.69 | 63.22 | 71.66 |
| CoCoOp [29] | 80.47 | 71.69 | 75.83 |
| VPT [10] | 80.17 | 73.60 | 76.47 |
| MaPLe [12] | 82.28 | 75.14 | 78.55 |
| Bloom | **83.24** | **76.87** | **79.68** |
|  | +0.96 | +1.73 | +1.13 |

**(b) ImageNet**

|  | Base | Novel | HM |
|---|---|---|---|
| CLIP [20] | 72.43 | 68.14 | 70.22 |
| CoOp [30] | 76.47 | 67.88 | 71.92 |
| CoCoOp [29] | 75.98 | 70.43 | 73.10 |
| VPT [10] | 75.70 | 69.10 | 72.25 |
| MaPLe [12] | 76.66 | 70.54 | 73.47 |
| Bloom | **77.15** | **71.25** | **74.08** |
|  | +0.49 | +0.71 | +0.61 |

**(c) Caltech101**

|  | Base | Novel | HM |
|---|---|---|---|
| CLIP [20] | 96.84 | 94.00 | 95.40 |
| CoOp [30] | 98.00 | 89.81 | 93.73 |
| CoCoOp [29] | 97.96 | 93.81 | 95.84 |
| VPT [10] | 97.73 | 93.10 | 95.36 |
| MaPLe [12] | 97.74 | 94.36 | 96.02 |
| Bloom | **98.60** | **96.28** | **97.43** |
|  | +0.86 | +1.92 | +1.41 |

**(d) OxfordPets**

|  | Base | Novel | HM |
|---|---|---|---|
| CLIP [20] | 91.17 | 97.26 | 94.12 |
| CoOp [30] | 93.67 | 95.29 | 94.47 |
| CoCoOp [29] | 95.20 | 97.69 | 96.43 |
| VPT [10] | 94.03 | 94.03 | 94.03 |
| MaPLe [12] | 95.43 | **97.76** | 96.58 |
| Bloom | **98.47** | 96.78 | **97.62** |
|  | +3.04 | -0.98 | +1.04 |

**(e) StanfordCars**

|  | Base | Novel | HM |
|---|---|---|---|
| CLIP [20] | 63.37 | 74.89 | 68.65 |
| CoOp [30] | **78.12** | 60.40 | 68.13 |
| CoCoOp [29] | 70.49 | 73.59 | 72.01 |
| VPT [10] | 69.83 | 74.23 | 71.97 |
| MaPLe [12] | 72.94 | 74.00 | 73.47 |
| Bloom | 74.34 | **76.64** | **75.47** |
|  | +1.40 | +2.64 | +2.00 |

**(f) Flowers102**

|  | Base | Novel | HM |
|---|---|---|---|
| CLIP [20] | 72.08 | **77.80** | 74.83 |
| CoOp [30] | **97.60** | 59.67 | 74.06 |
| CoCoOp [29] | 94.87 | 71.75 | 81.71 |
| VPT [10] | 91.50 | 70.70 | 79.77 |
| MaPLe [12] | 95.92 | 72.46 | 82.56 |
| Bloom | 96.04 | 75.36 | **84.45** |
|  | +0.08 | +2.90 | +1.89 |

**(g) Food101**

|  | Base | Novel | HM |
|---|---|---|---|
| CLIP [20] | 90.10 | 91.22 | 90.66 |
| CoOp [30] | 88.33 | 82.26 | 85.19 |
| CoCoOp [29] | 90.70 | 91.29 | 90.99 |
| VPT [10] | 90.07 | 91.13 | 90.60 |
| MaPLe [12] | 90.71 | 92.05 | 91.38 |
| Bloom | **92.54** | **94.46** | **93.49** |
|  | +1.83 | +2.41 | +2.11 |

**(h) FGVCAircraft**

|  | Base | Novel | HM |
|---|---|---|---|
| CLIP [20] | 27.19 | 36.29 | 31.09 |
| CoOp [30] | **40.44** | 22.30 | 28.75 |
| CoCoOp [29] | 33.41 | 23.71 | 27.74 |
| VPT [10] | 33.50 | 34.47 | 33.98 |
| MaPLe [12] | 37.44 | 35.61 | 36.50 |
| Bloom | 36.64 | **38.99** | **37.78** |
|  | -0.80 | +3.38 | +1.28 |

**(i) SUN397**

|  | Base | Novel | HM |
|---|---|---|---|
| CLIP [20] | 69.36 | 75.35 | 72.23 |
| CoOp [30] | 80.60 | 65.89 | 72.51 |
| CoCoOp [29] | 79.74 | 76.86 | 78.27 |
| VPT [10] | 78.33 | 77.57 | 77.95 |
| MaPLe [12] | 80.82 | 78.70 | 79.75 |
| Bloom | **82.85** | **79.33** | **81.05** |
|  | +2.03 | +0.63 | +1.30 |

**(j) DTD**

|  | Base | Novel | HM |
|---|---|---|---|
| CLIP [20] | 53.24 | **59.90** | 56.37 |
| CoOp [30] | 79.44 | 41.18 | 54.24 |
| CoCoOp [29] | 77.01 | 56.00 | 64.85 |
| VPT [10] | 77.27 | 54.77 | 64.10 |
| MaPLe [12] | 80.36 | 59.18 | 68.16 |
| Bloom | **82.16** | 59.83 | **69.24** |
|  | +1.80 | +0.65 | +1.08 |

**(k) EuroSAT**

|  | Base | Novel | HM |
|---|---|---|---|
| CLIP [20] | 56.48 | 64.05 | 60.03 |
| CoOp [30] | 92.19 | 54.74 | 68.69 |
| CoCoOp [29] | 87.49 | 60.04 | 71.21 |
| VPT [10] | 92.57 | 74.53 | 82.58 |
| MaPLe [12] | **94.07** | 73.23 | 82.35 |
| Bloom | 93.29 | **75.85** | **83.67** |
|  | +0.72 | +1.32 | +1.09 |

**(l) UCF101**

|  | Base | Novel | HM |
|---|---|---|---|
| CLIP [20] | 70.53 | 77.50 | 73.85 |
| CoOp [30] | **84.69** | 56.05 | 67.46 |
| CoCoOp [29] | 82.33 | 73.45 | 77.64 |
| VPT [10] | 81.33 | 76.00 | 78.58 |
| MaPLe [12] | 83.00 | 78.66 | 80.77 |
| Bloom | 83.52 | **80.90** | **82.19** |
|  | +0.52 | +2.24 | +1.42 |

Table 1: Comparison with state-of-the-art methods on base-to-novel generalization. The improvement of *Bloom* in green is relative to the state-of-the-art method with the highest HM.

where $[\cdot, \cdot, \cdot]$ denotes concatenate operation. Conversely, at the $i^{th}$ layer $\mathcal{L}_i$ within the text encoder, $M_l^i$ is first concatenated with the text embeddings $W_{i-1}$ obtained from the $(i-1)^{th}$ layer. This combined output is then fed into $\mathcal{L}_i$ to enable $V \rightarrow L$ interaction:

$$[\, W_i, \_\,] = \mathcal{L}_i([\, W_{i-1}, M_l^i \,]). \tag{12}$$

As a result, both $P_v$ and $P_l$ are updated by the language and vision branches simultaneously, thereby enhancing the overall flexibility of tuning process. In general, we apply *Bloom* to all 12 Transformer layers, resulting in a prompt depth of 12. We also investigate various configurations to evaluate the impact of prompt depth and placement position in experiments.

## 5 EXPERIMENTS

### 5.1 Experimental Protocol and Datasets

Building upon the foundation laid by previous research [12, 29, 30], we undertake a comprehensive evaluation of our method's generalization capabilities from three distinct perspectives.

| | Source | Target | | | | | | | | | | |
|---|---|---|---|---|---|---|---|---|---|---|---|---|
| | ImageNet | Caltech101 | OxfordPets | StanfordCars | Flowers102 | Food101 | Aircraft | SUN397 | DTD | EuroSAT | UCF101 | *Average* |
| CoOp [30] | **71.51** | 93.70 | 89.14 | 64.51 | 68.71 | 85.30 | 18.47 | 64.15 | 41.92 | 46.39 | 66.55 | 63.88 |
| CoCoOp [29] | 71.02 | **94.43** | 90.14 | 65.32 | 71.88 | 86.06 | 22.94 | 67.36 | 45.73 | 45.37 | 68.21 | 65.74 |
| VPT [10] | 69.43 | 93.53 | 89.73 | 64.43 | 68.50 | 84.77 | 24.17 | 66.53 | 45.57 | 36.23 | 65.93 | 63.94 |
| MaPLe [12] | 70.72 | 93.53 | 90.49 | 65.57 | 72.23 | 86.20 | **24.74** | 67.01 | 46.49 | 48.06 | 68.69 | 66.30 |
| Bloom | 70.58 | 93.83 | **90.79** | **66.13** | **72.80** | **86.71** | 24.58 | **67.46** | **46.87** | **50.32** | **69.17** | **66.87** |

**Table 2: Comparison of *Bloom* with state-of-the-art on cross-dataset evaluation.**

| | Source | Target | | | |
|---|---|---|---|---|---|
| | ImageNet | ImageNetV2 | ImageNet-S | ImageNet-A | ImageNet-R |
| CLIP [20] | 66.73 | 60.83 | 46.15 | 47.77 | 73.96 |
| CoOp [30] | **71.51** | 64.20 | 47.99 | 49.71 | 75.21 |
| CoCoOp [29] | 71.02 | 64.07 | 48.75 | 50.63 | 76.18 |
| VPT [10] | 69.43 | 62.80 | 48.03 | 45.90 | 75.83 |
| MaPLe [12] | 70.72 | 64.07 | 49.15 | 50.90 | 76.98 |
| Bloom | 70.58 | **64.50** | **50.37** | **51.56** | **77.13** |

**Table 3: Comparison of *Bloom* with existing approaches in cross-domain evaluation. *Bloom* outperforms existing methods on all *Target* datasets.**

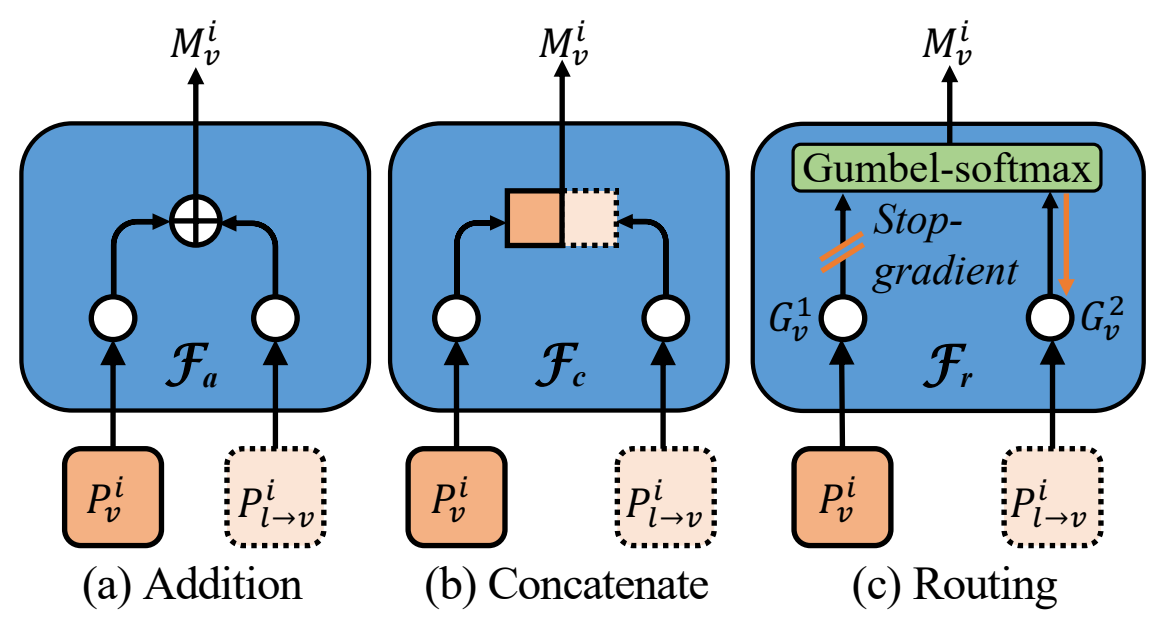

**Figure 4: Illustrations of 3 variants of fusion functions $\mathcal{F}$. We only present the $L \to V$ fusion process, while the $V \to L$ fusion procedure follows a similar approach.**

i) *Base-to-Novel Generalization:* Our evaluation of base-to-novel generalization encompasses 11 diverse image classification datasets. These include two generic-objects datasets, namely ImageNet [3] and Caltech101 [5]; five fine-grained datasets, specifically OxfordPets [19], StanfordCars [13], Flowers102 [18], Food101 [1], and FGVCAircraft [16]; a scene recognition dataset, SUN397 [27]; an action recognition dataset, UCF101 [23]; a texture dataset, DTD [2]; and a satellite image dataset, EuroSAT [7]. Each dataset's categories are bifurcated into "base" and "novel" classes. The "base" classes serve as the training and evaluation ground for base performance, while the "novel" classes are leveraged to assess the generalization to unseen classes in a zero-shot setting. We present the Top-1 accuracy for both the "base" and "novel" classes, along with their Harmonic Mean (HM).
ii) *Cross-dataset Generalization:* To ascertain the robustness and adaptability of our method across varied data distributions, we employ a zero-shot learning approach. This involves the direct transfer of the optimal model, initially trained on ImageNet, to the aforementioned 10 classification datasets. This rigorous testing process provides a comprehensive understanding of our method's effectiveness across various data landscapes.
iii) *Cross-domain Generalization:* Lastly, we scrutinize cross-domain generalization capabilities of *Bloom*. This is achieved by transferring the optimal model trained on ImageNet in a few-shot setting, to its four variants, namely, ImageNetV2 [22], ImageNetSketch [24], ImageNet-A [9], and ImageNet-R [8]. This final step of evaluation further solidifies our understanding of the method's adaptability and effectiveness across different domains.

## 5.2 Implementation Details

We train all models using a single Tesla V100 GPU for 5 epochs, employing a batch size of 4 and a learning rate of 0.0035 via the SGD optimizer. As the foundational model, we select the pre-trained ViT-B/16 CLIP model, in which the dimensions of the visual and language representations, $d_v$ and $d_l$, are set to 768 and 512, respectively. We report the Top-1 accuracy for both base and novel classes, as well as their harmonic mean (HM), averaged over three runs with random seeds $\in \{1, 2, 3\}$. The language prompt $P_l$ is initialized using a static prompt, such as "`a photo of a [CLS]`.". It is worth noting that the model trained on all 1,000 classes of ImageNet, which is utilized for evaluating cross-dataset generalization, undergoes training for only 2 epochs with a learning rate of 0.0026.

## 5.3 Base-to-novel Generalization

In this study, we investigate the base-to-novel generalization capabilities of *Bloom* across 11 image classification datasets. We employ a **basic setting** in which the prompt depth is set to 12, signifying the application of *Bloom* to every layer. The prompt length $L$ is configured to 2, and the training is conducted in a 16-shot setting, which entails randomly selecting 16 samples for each category. Additionally, the middle dimension $d_m$ within the projection functions is set to 64. The results are presented in Tab. 1. We benchmark the performance of our approach against state-of-the-art prompt learning methods, which include CoOp [30], CoCoOp [29], VPT [10], MaPLe [12], and the zero-shot CLIP model utilizing a hand-crafted prompt. Specifically, with respect to the harmonic mean (HM) of

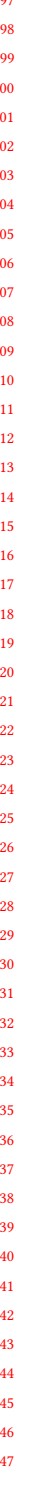

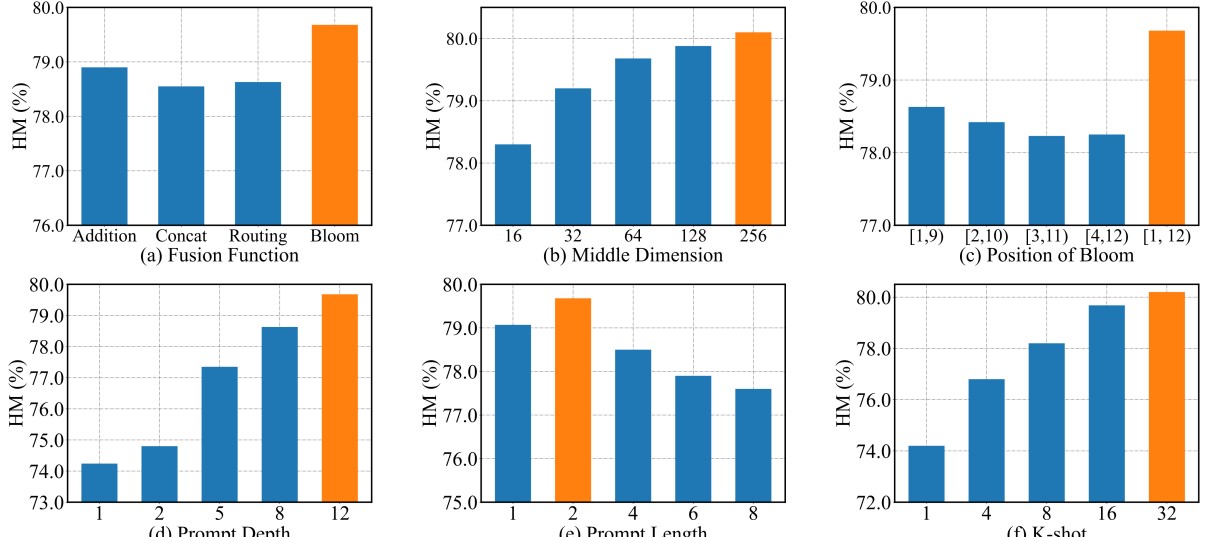

**Figure 5: Ablation study to investigate the impact of *Bloom*'s crucial designs and hyper-parameters on the ImageNet dataset within the context of a base-to-novel training. The orange bar indicates the optimal setting.**

the accuracy for both "base" and "novel" classes, *Bloom* surpasses the state-of-the-art methods on all of 11 datasets, with significant improvements in terms of "base" accuracy (+0.96%), "novel" generalization (+1.73%), and their HM (+1.13%), averaged across the 11 datasets. Although *Bloom*'s performance is sub-optimal in some instances, such as a -0.80% decrease in "Base" accuracy on the FGVCAircraft dataset, it demonstrates significant improvements in "Novel" classes and attains the best Harmonic Mean (HM) compared to state-of-the-art methods. This highlights the method's ability to adapt and excel in various scenarios, despite occasional shortcomings.

## 5.4 Cross-dataset Generalization

We investigate the cross-dataset generalization capabilities of *Bloom* by utilizing ImageNet as the source dataset and employing the remaining 10 datasets as target datasets. Following the recommendations from prior research [12], we adopt a shallow and slim prompt configuration, setting the prompt depth to 2 and $L = 1$. The results, as depicted in Tab. 2, reveal that our *Bloom* surpasses the state-of-the-art methods on 8 out of 10 datasets in terms of zero-shot accuracy. On average, *Bloom* achieves an improvement of +0.57%, signifying enhanced generalization performance in comparison to existing state-of-the-art approaches.

## 5.5 Cross-domain Generalization

Analogous to the cross-dataset generalization evaluation, we directly transfer the source model trained on ImageNet to four cross-domain datasets. As illustrated in Tab. 3, the *Bloom* outperforms state-of-the-art methods on all of the tested datasets. This result underscores the superior generalization capabilities of *Bloom* in addressing domain shifts, highlighting its effectiveness in adapting to diverse and challenging scenarios.

## 5.6 Ablation Study

We carry out an ablation study, delving into the effects of crucial designs and hyper-parameters on the ImageNet dataset within the context of a base-to-novel training setting.

**The Formulation of Fusion Function.** To validate the crucial role of the adaptive cross-modal fusion function, we evaluate 3 additional fusion functions that exhibit lower flexibility compared to *Bloom*: i) *Addition* ($\mathcal{F}_a$), which entails the addition of corresponding elements from the two prompts. ii) *Concatenate* ($\mathcal{F}_c$), which involves concatenating the two prompts. iii) *Routing* ($\mathcal{F}_r$), which is built upon $\mathcal{F}_g$ (Eq. 10). We propose a routing function that replaces $\mathcal{G}_v$ and $\mathcal{G}_l$ with two binarized gate vectors via a differentiable Gumbel-softmax function. Consequently, the routing strategy permits only uni-modal prompts across each gate, while preserving the potential for cross-modal interaction from a macroscopic perspective. An illustration of the variations is provided in Fig. 4. We employ the same **basic setting** across all variations of the fusion function, with the results depicted in Fig. 5 (a). The superior performance of *Bloom* is evident, thus corroborating the notion that a more flexible cross-modal interaction leads to a more proper alignment.

**Middle Dimension.** The middle dimension $d_m$ within projection functions plays a pivotal role in achieving a better trade-off between enhanced performance and minimal additional learnable parameters. In the **basic setting**, we set $d_m$ to 16, 32, 64, 128, 256, respectively, while maintaining other hyper-parameters constant. As illustrated in Fig. 5 (b), the HM on ImageNet increases as $d_m$ becomes larger with diminishing marginal effects. When setting $d_m$ to 256, the HM surpasses 80%, but the learnable parameters double, contradicting the desired trade-off objective.

**Where to insert Bloom?** In this study, we examine the impact of varying the insertion position of *Bloom*. We establish 5 distinct intervals, namely [1, 9), [2, 10), [3, 11), [4, 12), [1, 12), to determine the placement of *Bloom*, where [$a$, $b$) signifies that the

| Method | Base | Novel | HM |
|---|---|---|---|
| Visual→Language (V→L) only | 76.28 | 70.20 | 73.11 |
| Language →Visual (L→V) only | 76.53 | 69.96 | 73.09 |
| Visual ↔ Language (*Bloom*) | **77.15** | **71.25** | **74.08** |

**Table 4: The effect of cross-modal fusion direction.**

| Method | Params | Params %CLIP | HM |
|---|---|---|---|
| CoOp [30] | 2.048K | **0.002** | 71.66 |
| CoCoOp [29] | 35.36K | 0.03 | 75.83 |
| VPT [10] | 73.73K | 0.06 | 72.25 |
| MaPLe [12] | 3.55M | 2.85 | 78.55 |
| Bloom | 2.24M | 1.80 | **79.68** |

**Table 5: Comparison of the number of additional trainable parameters and HM (%) on ImageNet dataset. Our proposed *Bloom* yields the best trade-off.**

prompt is inserted into $(b - a)$ consecutive layers, spanning from the $a^{th}$ layer to the $(b - 1)^{th}$ layer. The other hyper-parameters in **basic setting** are constant. The results are depicted in Fig. 5 (c). Our findings indicate that as the insertion position shifts further toward the latter layers, there is a gradual decline in performance. Consequently, we deduce the first principle for the usage of *Bloom* that *the earlier the mutual interaction, the better the alignment*. This implies that the *Bloom* plays a more crucial role during the initial stages of CLIP. Incorporating cross-modal fusion at an earlier point in the model leads to improved alignment.

**Prompt Depth.** In this study, we investigate the influence of prompt depth by adjusting it to {1, 2, 5, 8, 12}, while remain the other hyper-parameters in **basic setting** constant. The results are presented in Fig. 5 (d). Our analysis reveals a positive correlation between the prompt depth and the HM, indicating an improvement in performance as the depth increases. Consequently, we deduce the second principle for the application of *Bloom* that *the deeper the mutual interaction, the better the alignment*. In particular, a more effective *Bloom* implementation, which integrates the two modalities across multiple layers, promotes enhanced cross-modal alignment. This, in turn, results in an optimized balance between performance on "base" classes and generalization capabilities on "novel" classes.

**Prompt Length.** In our subsequent analysis, we examine the impact of the prompt length $L$ in Eq. 6 by adjusting it to {1, 2, 4, 6, 8}. The other hyper-parameters are identical to **basic setting**. As shown in Fig. 5 (e), our findings indicate that, in the majority of cases, an increase in prompt length results in a decline in the HM. This suggests that employing a larger prompt length may not be optimal. We postulate that the *Bloom* with an extended length exacerbates the challenges associated with cross-modal alignment due to the absence of adequate prior information.

**The Effect of Data Scaling for Few-shot Training.** In our ongoing investigation, we explore the impact of *data scaling* on model performance by training the *Bloom* within a $k$-shot learning framework, where $k$ encompasses {1, 4, 8, 16, 32}. The corresponding

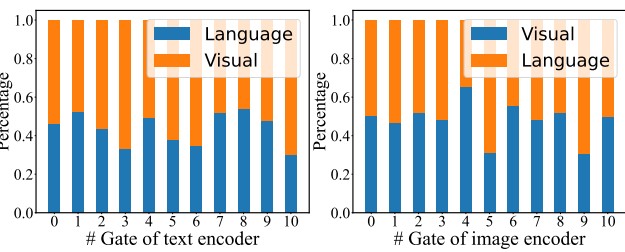

**Figure 6: Visualization of gate vectors within fusion functions, illustrating Bloom's search decisions regarding cross-modal information at various layers.**

results are illustrated in Fig. 5 (f). Our analysis reveals a favorable effect of *data scaling*, as evidenced by the increase in the HM when more few-shot samples are incorporated. However, it is worth noting that the benefits derived from augmenting the training dataset exhibit diminishing marginal returns. As the number of training samples increases, the incremental gains in performance become progressively smaller.

## 5.7 Discussions

**The Effect of Cross-modal Fusion Direction.** We conduct a comparative analysis of the performance of two distinct uni-directional cross-modal fusion strategies, designated as V→L and L→V, and *Bloom*, denoted as V↔L. The comparison, as illustrated in Tab. 4, reveals that *Bloom* outperforms both uni-directional strategies. The superiority of *Bloom* can be attributed to its ability to leverage the mutual information across two modalities, thereby facilitating a more comprehensive and proper alignment. In contrast, uni-directional strategies may not fully capitalize on the potential benefits of integrating information from both modalities, resulting in sub-optimal performance.

**The Trade-off between Performance and Parameters.** In Tab. 5, we compare the HM on ImageNet and additional trainable parameters of *Bloom* with state-of-the-art approaches, including CoOp [30], CoCoOp [29], VPT [10], and MaPLe [12]. The results demonstrate that *Bloom* achieves a superior trade-off, requiring only 1.8% of additional CLIP parameters while attaining a 79.68% HM.

**The Search Decision of Gate Vectors.** We visualize the gate vectors (after softmax) within the fusion functions of both the vision and language branches in Fig. 6. This intuitively showcases *Bloom*'s varying requirements for mutual information from the two different modalities across distinct layers. Moreover, this observation corroborates our hypothesis, suggesting that providing prompt learning with greater autonomy positively impacts its performance.

## 6 CONCLUSIONS

In this paper, we introduce a new paradigm, termed *Bloom*, for prompt learning community. *Bloom* exhibits more flexibility for cross-modal interaction via a bilateral cross-modal fusion framework and an adaptive fusion function ensuring *Bloom* to search optimal combination of interaction information, ultimately resulting in new state-of-the-art performance in terms of base-to-novel generalization, cross-dataset, and cross-domain generalization.

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
