# OpenReview forum: "Bilateral Adaptive Cross-Modal Fusion Prompt Learning for CLIP"
_acmmm.org/ACMMM/2024/Conference — MM2024 Poster_

### Official Review · Reviewer_nGbT · 2024-04-30

**Rating:** 5
**Confidence:** 3

**Summary:**

This paper introduces a new paradigm for prompt learning, termed Bloom. Bloom exhibits more flexibility for cross-modal interaction via a bilateral cross-modal fusion framework and an adaptive fusion function ensuring Bloom to search optimal combination of interaction information, ultimately resulting in new SOTA performance in terms of base-to-novel generalization, cross-dataset, and cross-domain generalization.

**Strengths:**

1. The idea is clear, simple, yet effective.
2. The writing is clear and logical.
3. The experiments are solid and sufficient, demonstrating the generalization ability of the proposed method.

**Limitations:**

1. The training GPU memory usage should be discussed in the discussion of trade-off.
2. Fig.6 is hard to understand.
3. Though on different tasks, the idea of bi-directional projection of prompt fusion was proposed in PMF [1], which reduces the novelty of this paper.

[1] Yaowei Li, Ruijie Quan, Linchao Zhu, and Yi Yang. 2023. Efficient Multimodal Fusion via Interactive Prompting. In Proceedings of the IEEE/CVF Conference on Computer Vision and Pattern Recognition. 2604–2613.

**Suitability:**

3

---

### Official Review · Reviewer_woSY · 2024-05-09

**Rating:** 5
**Confidence:** 2

**Summary:**

The paper presents Bilateral Adaptive Cross-Modal Fusion Prompt Learning (Bloom), a novel approach for improving the adaptation of the CLIP model to downstream tasks. Bloom enhances cross-modal alignment by introducing bi-directional fusion of visual and textual information across both the image and text encoders. It employs adaptive fusion functions that automatically determine the optimal integration of cross-modal data at each layer, leading to a more comprehensive alignment. Experimentally, Bloom surpasses existing state-of-the-art methods in three critical areas: base-to-novel generalization, cross-dataset, and cross-domain evaluations.  The work validates the importance of early and deep cross-modal interactions for better alignment and generalization, and visualizes gate vectors to showcase Bloom's dynamic adjustment of cross-modal information requirements. Overall, Bloom represents a significant step forward in prompt learning for CLIP, offering a more flexible and efficient way to leverage cross-modal representations in diverse applications.

**Strengths:**

1)	Importance of key insights: the paper proposes an innovative Bloom approach that addresses the lack of adaptability of existing CLIP models through bi-directional adaptive cross-modal fusion;.
2)	Experimental or theoretical validation: through exhaustive experimental design, including cross-dataset, cross-domain and base-to-new generalisation evaluation, as well as meticulous ablation studies, the effectiveness of Bloom's approach is empirically demonstrated, proving that more flexible cross-modal interactions enhance the model's generalisation capability;.
3)	Writing quality: the article is well-organised and logical, detailing the principles of the method, experimental design and analysis of the results, as well as being rich in graphs and charts.
4)	Data contribution: comprehensive evaluation data on 15 image classification datasets are provided, demonstrating the significant advantages of Bloom over existing top methods.

**Limitations:**

1)	The paper discusses less about the theoretical basis of the proposed two-way fusion function and adaptive strategy；
2)	Lack of sufficient implementation details and hyperparameter settings in the paper；

**Suitability:**

3

---

### Official Review · Reviewer_R1Xo · 2024-05-25

**Rating:** 3
**Confidence:** 3

**Summary:**

This paper proposes Bilateral Adaptive Cross-Modal Fusion Prompt Learning (Bloom), a new paradigm for the CLIP prompt learning community. Bloom enhances cross-modal interaction through a bilateral fusion framework and an adaptive fusion function, ensuring optimal interaction combinations. It features two main enhancements: projection functions for bi-directional modality transformation and fusion functions for layer interaction within image and text encoders, and an adaptive method to find the best combination of cross-modal information at each layer. Extensive experiments on 15 image classification datasets demonstrate its effectiveness.

**Strengths:**

1. This paper is well-written.
2. The organization of the paper is logical.

**Limitations:**

1. From my point of view, the proposed bilateral adaptive cross-modal fusion prompt learning is more like a variant of the MaPLe and similar to its architecture, this may weaken the novelty of the proposed method. The authors are expected to further explain and refine their motivation and contribution.
2. It is better to provide the image embeddings of the proposed method via t-SNE or UMAP and compare it with previous methods such as MaPLE and Co-CoOp.

**Suitability:**

3

---

### Official Review · Reviewer_pwKE · 2024-05-29

**Rating:** 4
**Confidence:** 4

**Summary:**

This article proposes a simple  Bilateral Adaptive Cross-Modal Fusion method. It implements prompt tuning of CLIP by implanting learnable vectors and projection modules between the two encoders of CLIP, and achieves good results.

**Strengths:**

1. This paper is clearly written and easy to understand.
2. The method is simple and effective, and its sufficient generalization ability has been verified on a variety of tasks.
3. A large number of experiments fully explored the effectiveness and versatility of this method.

**Limitations:**

1. Although it does not use complex distillation or ensemble operations, the performance of this article does not exceed prompt tuning methods such as PromptSRC. Can you try to add your method on PromptSRC?
2. This article seems to be relatively uninnovative. It is just a simple replacement of the fusion module on methods such as MaPLe.
3. The H values in Table 4 look different from those in the other charts, where H usually reaches 79.68, and only 74.08 in Table 4. It is expected that you can align the experimental settings.
4. Although this is not necessary, it is recommended to add time and space metrics such as FLOP and Memory to enrich Bloom's efficiency comparison with other methods.
5. I would like to know why Bloom has fewer parameters than MaPLe? Bloom implanted Prompt at more layers and learned bidirectional mapping, which appears to have more parameters.

**Suitability:**

3

---

### Meta-Review · Area_Chair_XKLS · 2024-07-03

**Recommendation:** Accept (Poster)
**Confidence:** 5

**Metareview:**

This paper introduces a cross-modal fusion prompt learning method for CLIP. The original reviews are two weak accept, one borderline accept, and one borderline reject. The main concerns center around the novelty of the proposed method. The rebuttal successfully addressed some of the concerns. The authors are encouraged to incorporate all the comments in the final version.